# Protofibrillar Amyloid Beta Modulation of Recombinant hCaV2.2 (N-Type) Voltage-Gated Channels

**DOI:** 10.3390/ph15121459

**Published:** 2022-11-24

**Authors:** Eleni Kaisis, Laura J. Thei, Gary J. Stephens, Mark L. Dallas

**Affiliations:** School of Pharmacy, University of Reading, Reading RG6 6UB, UK

**Keywords:** ion channel, amyloid beta, voltage-gated calcium channel, neurodegeneration, Alzheimer’s disease

## Abstract

Cav2.2 channels are key regulators of presynaptic Ca^2+^ influx and their dysfunction and/or aberrant regulation has been implicated in many disease states; however, the nature of their involvement in Alzheimer’s disease (AD) is less clear. In this short communication, we show that recombinant hCav2.2/b_1b_/a_2_d_1_ channels are modulated by human synthetic AD-related protofibrillar amyloid beta Ab_1-42_ peptides. Structural studies revealed a time-dependent increase in protofibril length, with the majority of protofibrils less than 100 nm at 24 h, while at 48 h, the majority were longer than 100 nm. Cav2.2 modulation by Ab_1-42_ was different between a ‘low’ (100 nM) and ‘high’ (1 µM) concentration in terms of distinct effects on individual biophysical parameters. A concentration of 100 nM Ab_1-42_ caused no significant changes in the measured biophysical properties of Cav2.2 currents. In contrast, 1 µM Ab_1-42_ caused an inhibitory decrease in the current density (pA/pF) and maximum conductance (Gmax), and a depolarizing shift in the slope factor (k). These data highlight a differential modulation of Cav2.2 channels by the Ab_1-42_ peptide. Discrete changes in the presynaptic Ca^2+^ flux have been reported to occur at an early stage of AD; therefore, this study reveals a potential mechanistic link between amyloid accumulation and Ca_v_2.2 channel modulation.

## 1. Introduction

The amyloid beta (Ab) peptide has been implicated in the pathogenesis of AD, with the proposal that lowering Ab levels is beneficial to brain health [1,2]. While the evidence supporting the involvement of Ab in the disease state is robust, there is a growing interest in the ‘normal’ physiological role that Ab plays [3]. Most notably, the removal of endogenous Ab using genetic and immunological methodology has a wide range of adverse effects (see [4]). More targeted approaches have revealed that Ab contributes to physiology through its antimicrobial activity, tumour suppression, neuroprotection and regulation of synaptic plasticity [5]. These functions require the interaction of Ab with a wide array of molecular targets. One superfamily of molecular targets are ion channels, which underpin a compendium of biological processes. While the Ab-mediated dysfunction of ion channels has been implicated in disease progression, there is also evidence to support the modulation of ion channel function in maintaining a healthy CNS [6,7]. Overall, Ab is likely to play a unique role in modulating brain ionic homeostasis via the regulation of ion channel activity.

Voltage-gated calcium channels (VGCCs) are pivotal molecular entities in regulating neuronal Ca^2+^ homeostasis [8]. This has led to several studies investigating the role of Ab in influencing the activity of VGCCs [9,10,11,12,13]. Preclinical studies have predominantly focused on the dihydropyridine-sensitive L-type VGCCs [12,14,15,16], demonstrating increases in Ca^2+^ influx through L-type channels mediated by a range of Ab assemblies. In turn, clinical studies have looked to exploit existing antihypertensive calcium channel blockers as novel AD therapeutics; however, these data are conflicting. Studies looking at patients with a history of taking antihypertensives and subsequently developing dementia provide a mixed picture [17,18,19]. Moreover, a more targeted intervention using nilvadipine, which inhibits both L-type VGCCs and spleen tyrosine kinase activity, failed to demonstrate efficacy in the management of existing dementia in terms of modulating measures of cognition [20]. Ca_V_2.2 (N-type) VGCCs have been somewhat overlooked for their potential contribution to amyloid pathology. Ca_V_2.2 VGCCs are presynaptic, ω-conotoxin-sensitive and G-protein-modulated channels that are known targets for pain therapeutics [21]. The link between dementia and pain is also well established (see [22]); recent research has demonstrated a temporal profile and indicates that pain could be classified as a prodromal symptom for dementia [23]. Therefore, further investigation of the Ab modulation of N-type channels is merited. This study demonstrates that the recombinant human Ca_V_2.2/b_1b_/a_2_d channel complex is a target for human synthetic protofibrillar Ab_1-42_ and that there is a concentration relationship for the effects on discrete biophysical properties. These findings have implications for the management of Ca^2+^ influx in AD in particular, given the presynaptic localisation of Ca_V_2.2 subunits [24,25].

## 2. Results

### 2.1. Protofibrillar Assembly of Amyloid

To verify the Aβ_1-42_ form used in our functional experiments, an analysis of peptide aggregation was undertaken. The transmission electron microscope images revealed that at 0 h, the peptide was largely non-aggregated and was observed as Aβ_1-42_ globules. At this time point, no fibril structures were evident (Figure 1a). After 24 h of incubation, the formation of small protofibrils (defined as <100 nm in length) could be detected (Figure 1b) and by 48 h, aggregation into longer protofibrils was observed (Figure 1c). In comparison, no protofibril structures were observed when using the reverse sequence peptide, Aβ_1-42_ (Figure 1d). A fibril length analysis revealed a time dependency of the formation of Aβ_1-42_ protofibrils, with a 41.4 ± 7.1% (n = 25) increase in protofibril length from a 24 to 48 h incubation. After 24 h of incubation, the majority (68.0%) of the amyloid structures analysed were protofibrils of less than 100 nm in length (average length of 93.2 ± 3.9 nm; n = 25). By contrast, at 48 h the majority (89.3%) of the protofibrils were greater than 100 nm in length (average length of 131.8 ± 5.8 nm; n = 28) (Figure 1e).

### 2.2. Protofibrillar Amyloid Beta Modulation of Ca_V_2.2 Currents

The effects of protofibrillar Aβ_1-42_ on Cav2.2 currents was investigated via population experiments using HEK293 cells expressing the Ca_V_2.2/β_1b_/α_2_δ_1_ complex, where cells were exposed to either synthetic Aβ_1-42_ or PBS vehicle control for 24 h prior to recordings. A multi-step voltage protocol was applied to Ca_V_2.2 stably transfected HEK293 cells to elicit the current density/voltage relationships for the experimental groupings. The 100 nM protofibrillar Aβ_1-42_ showed no significant inhibition of the Cav2.2 current density compared to the vehicle control (control: −46.4 ± 15.1 pA/pF vs. amyloid treated: −31.7 ± 7.8 pA/pF, *p* = 0.36, n = 8–11; Figure 2 and Figure 3a,b).

In addition, HEK293 cells expressing the Ca_V_2.2/β_1b_/α_2_δ_1_ complex treated with 100 nM protofibrillar Aβ_1-42_ for 24 h had no effect on the measured biophysical characteristics (Figure 3c–f).

The effects of a higher concentration of protofibrillar Aβ_1-42_ on Ca_V_2.2 were investigated (Figure 4 and Figure 5). A concentration of 1 µM protofibrillar Aβ_1-42_ (24 h) caused a significant reduction in the current density (vehicle control: −47.1 ± 12.5 pA/pF vs. amyloid treated: −17.7 ± 5.4 pA/pF, *p* = 0.017, n = 8–11; Figure 4 and Figure 5a,b). Correspondingly, a significant reduction in G_max_ was also observed (control: 1.3 ± 0.2 nS/pF vs. amyloid treated: 0.6 ± 0.1 nS/pF, *p* = 0., n = 8–11; Figure 4 and Figure 5f). The slope factor (k) was also altered after exposure to 1 µM protofibrillar Aβ_1-42_ (vehicle control: −4.5 ± 0.4 vs. amyloid treated: −7.3 ± 0.7, *p* = 0.003, n = 8–11; Figure 5e). The other biophysical properties examined (V_0.5_ and reversal potential) remained unaffected (Figure 5c–d).

## 3. Discussion

This short communication identifies protofibrillar Aβ, formed using 1% ammonium hydroxide [26], as an important modulator of recombinant Ca_V_2.2, β_1b_ and α_2_δ subunits containing VGCC complexes. The use and verification of protofibrillar Aβ is vital in elucidating the functional effects of disease-relevant Aβ species. It is evident that a lack of complete biophysical characterisation of Cav2.2 channels with potentially different subunit compositions may lead to a diverse array of modulatory affects and some apparently contradictory reports in the literature (see [9,11]). Ca_V_2.2 channels carrying N-type currents are predominantly involved at presynaptic loci [25]. For example, in the hippocampus, evidence supports a role for Ca_V_2.2 in inhibitory interneuron neurotransmission [27] and the development of pyramidal neuron synapses [28]. The non-neuronal expression of Cav2.2 has been reported [29,30]; however, there is a lack of functional data to indicate a role for a fully conducting channel. This has led to VGCCs being investigated as molecular targets for a range of neurological diseases, including AD. Aβ has been linked to the modulation of a host of VGCCs and is reported to affect channel function across different physiological levels (e.g., transcription vs. translation). At the expression level, the Aβ_25-35_ fragment has been reported to up-regulate Cav1.2, Cav2.1 and Cav2.2 mRNA and proteins in a time-dependent manner [31]. This is seen as central to facilitating neurotoxicity through an increased Ca^2+^ influx associated with elevated levels of Aβ. Another potential mechanism by which Aβ modulates VGCCs is through subunit/protein–protein interactions. Here, evidence supports the Aβ_25-35_ modulation of Ca_V_1.2 at the mRNA, protein and functional levels [32]; in addition, a role of the β3 subunit was proposed to mediate changes in the functional activity of both Cav1.2- and Cav1.3-containing channels. Aβ has also been reported to modulate VGCC function through the modulation of biophysical properties in other studies. An Aβ-mediated reduction in Ca^2+^ influx was reported to occur via a reduction in P/Q-type current amplitude and conductance, with the slope factor and V_50_ remaining unaltered [11]. By contrast, Mezler et al. (2012) showed that the treatment of *Xenopus laevis* oocytes with oligomeric synthetic Aβ_1-42_ peptides increased the amplitude and caused a hyperpolarized shift in the activation of recombinant Cav2.1 channels [9]. Hermann and colleagues also reported that Aβ_1-42_ globulomers not only modulated Cav2.1 kinetics, but also increased the activity of Cav2.2 channels via an increase in amplitude and a shift of V_50_ towards more hyperpolarized values [10]. Such differences may be attributed to different forms/aggregation states and/or concentrations of Aβ used. Here, we have looked at a distinct disease-relevant species of Aβ (protofibrillar) and revealed its effects on specific Cav2.2 kinetic parameters based on the concentration of the Aβ assembly. Lower concentrations (100 nM) showed no modulation of Cav2.2 currents, while higher concentrations (1 µM) caused significant changes in the Cav2.2 biophysical fingerprint. Here, reductions in current density and maximal conductance were reported alongside a change in the slope factor (k). Thus, Ab had inhibitory effects on Ca^2+^ currents. The slope factor k represents the change in membrane potential required to cause an e-fold change in the activation of open channels. A reported 5 mV hyperpolarising shift would lead to a decrease in the voltage sensitivity of the channel, such that larger changes in voltage were required to cause a similar level of channel activation. This change in the voltage sensitivity of the Cav2.2 channel could be the result of post-translational modulation, which has been reported for other Cav channels in neurodegenerative disorders [33].

While protofibrillar Ab is often associated with neuronal toxicity [34], our data indicate an inhibitory modulation of Cav2.2, which may relate to pain symptoms that manifest in AD rather than overt toxicity. Pain has previously been linked to AD, but common signalling pathways have not robustly been identified. For example, chronic pain has been reported to accelerate AD pathology and initiate cognitive decline earlier in transgenic mice [35]; this rapid onset was suggested to be due to changes in NMDA receptor subunit functionality within the hippocampus. In human studies, there is mixed evidence as to the role of chronic pain and the subsequent development of dementia. Those patients reporting higher pain scores were observed to have a higher probability of developing dementia, while others have indicated no relationship between the intensity of pain and a dementia diagnosis [36,37,38]. One established mechanism for nociception signalling is through presynaptic Ca_V_2.2 channels, which has led to the approval of ω-conotoxin MVIIA as the peptide drug, ziconotide, for chronic severe pain [39]. Therefore, further research is merited to better understand the role of Cav2.2 channels in AD pathology and pain. This should include consideration of neuronal and non-neuronal cells, with evidence indicating that genetic manipulation of Cav2.2 within the microglia can impact pain signalling [30]. A breakdown of Ca^2+^ homeostatic mechanisms is implicated in Aβ-induced neurodegeneration [40], and there is evidence linking disruption in Ca^2+^ signalling with the development of different pain states [41,42]. Therefore, understanding the individual molecular targets and their temporal involvement in pathology will aid drug discovery and support the use of available pharmacological tools in managing individuals. This will improve strategies for modulating VGCC function, which may be beneficial during the early synaptic dysfunctional stages of AD progression [43].

## 4. Materials and Methods

### 4.1. Cell Culture

Stably transfected HEK293 cells expressing human Ca_V_2.2 calcium channels (α_1B_ (Ca_v_2.2), β_1b_ and α_2_δ_1_ subunits) [44] were cultured in minimum essential medium (MEM; Life Technologies, Loughborough, UK) supplemented with 2 mM L-glutamine (Life Technologies, Loughborough, UK), 0.5 mg/mL geneticin (G418; Sigma Aldrich, Gillingham, UK), 10% FBS (Life Technologies, Loughborough, UK) and 1% penicillin/streptomycin (Life Technologies, Loughborough, UK). 

### 4.2. Aβ Assembly and Characterisation

Human synthetic Aβ_1-42_ peptides (Abcam, Cambridge, UK: Aβ_1-42_ #ab120301) or Aβ_42-1_ peptides (Abcam, Cambridge, UK: Aβ_42-1_ # ab120481) were dissolved in 1% ammonium hydroxide (NH_4_OH; Sigma Aldrich, Gillingham, UK) in PBS, taking into consideration the peptide content, to obtain a 1 mg peptide film as previously described [26]. Following this, Aβ was left at room temperature for 30 min and then placed in a speed-vac until a peptide film was obtained (~20 min). The peptide film was then dissolved into a 5 mM Aβ stock solution using 10 mM sodium hydroxide in dH_2_O (NaOH; Sigma Aldrich, Gillingham, UK). Aβ was sonicated for 10 min to prime the peptide for aggregation. Both the peptide film and the 5 mM Aβ stock were stored at −20 °C until use.

Prior to experimental use, the 5 mM stock was sonicated for a further 10 min and then further diluted into a 100 μM usable stock over a 48 h incubation period using a two-step protocol. Firstly, the 5 mM stock was diluted into 400 μM using PBS and 2% sodium dodecyl sulphate (SDS; Invitrogen, Loughborough, UK), and then incubated over 24 h. Next, 400 μM Aβ was diluted into 100 μM usable stock using PBS and incubated for another 24 h. For experimental purposes, 100 μM Aβ_1-42_ or Aβ_42-1_ was further diluted in media to achieve various working concentrations that the cells were then exposed to or that were processed for transmission electron microscopy.

### 4.3. Transmission Electron Microscopy

The aggregation processes of the Aβ_1-42_ preparations were investigated using transmission electron microscopy (TEM; Jeol 2100 Plus, 200 kV). Firstly, the Aβ preparations (30 µL) were loaded onto 300-mesh-sized carbon grids (HC300Cu films, EM Resolutions, Sheffield, UK). The grids were then stained with 1–2% *w*/*w* uranyl acetate (Agar, Stansted, UK) in dH_2_O (~1 min) for visualisation purposes. The grids were then washed with dH_2_O to remove excess stain (for ~1 min) and then left to dry (~5 min) prior to use. The image analysis, via measuring protofibril length, was achieved using Image J.

### 4.4. Electrophysiology

Glass electrodes were filled with an intracellular solution composed of: 120 mM CsCl; 20 mM TEA; 10 mM EGTA; 2 mM MgCl_2_; 10 mM HEPES; and 2 mM ATP (pH of 7.2 using CsOH). The extracellular solution was 95 mM NaCl (ThermoFisher Scientific, Loughborough, UK); 10 mM BaCl_2_; 2 mM MgCl_2_; 10 mM HEPES; 10 mM glucose; and 5 mM TEA (pH of 7.4 using NaOH). To study the VGCC currents, a multi-step protocol was applied from a holding potential of −70 mV, in 10 mV increments from −70 to +50 mV (for 200 ms). Data from multi-step recordings were used to plot the current density/voltage relationships. A single-step protocol to +10 mV from a holding potential of −70 mV was also applied. The extracellular solution was applied via a continuous tube-perfusion system. Prior to investigating the VGCC currents, cells were treated for 24 h with Aβ_1-42_ or the vehicle control (PBS).

All electrophysiology experiments were carried out at room temperature (~21 °C). Whole-cell patch clamp recordings were acquired at 10 kHz and low-pass-filtered at 2 kHz using an Axopatch 200B (Axon Instruments) and Digidata 1440A (Axon Instruments) using Clampex10.3 software. Electrophysiology recordings were analysed using Clampfit10.3.

### 4.5. Statistical Analysis

Statistical analyses and the plotting of graphs were performed using GraphPad Prism 7.0. For statistical analyses, all data were firstly tested for normality (Shapiro–Wilk) and then examined for outliers (Grubb’s), which were excluded from the analysis. Data were plotted as an average ± SEM with the number of biological replicates (n) as stated. Values of *p* > 0.05 were considered significant.

For the Aβ protofibril length analysis, one-way ANOVA followed by Tukey’s post hoc test was used.

For electrophysiology experiments, the current density/V relationships were first fitted with a modified Boltzmann fit equation (Y = (G_max_x(X − V_rev_))/(1 + exp((−(X − V_50_)/k)) [45], where G_max_ = maximum conductance; V_rev_ = reversal potential; V_50_ = half activation voltage; and k = slope factor. Data obtained from the Boltzmann fit analysis were then subjected to a two-tailed unpaired *t*-test [40].

## Figures and Tables

**Figure 1 pharmaceuticals-15-01459-f001:**
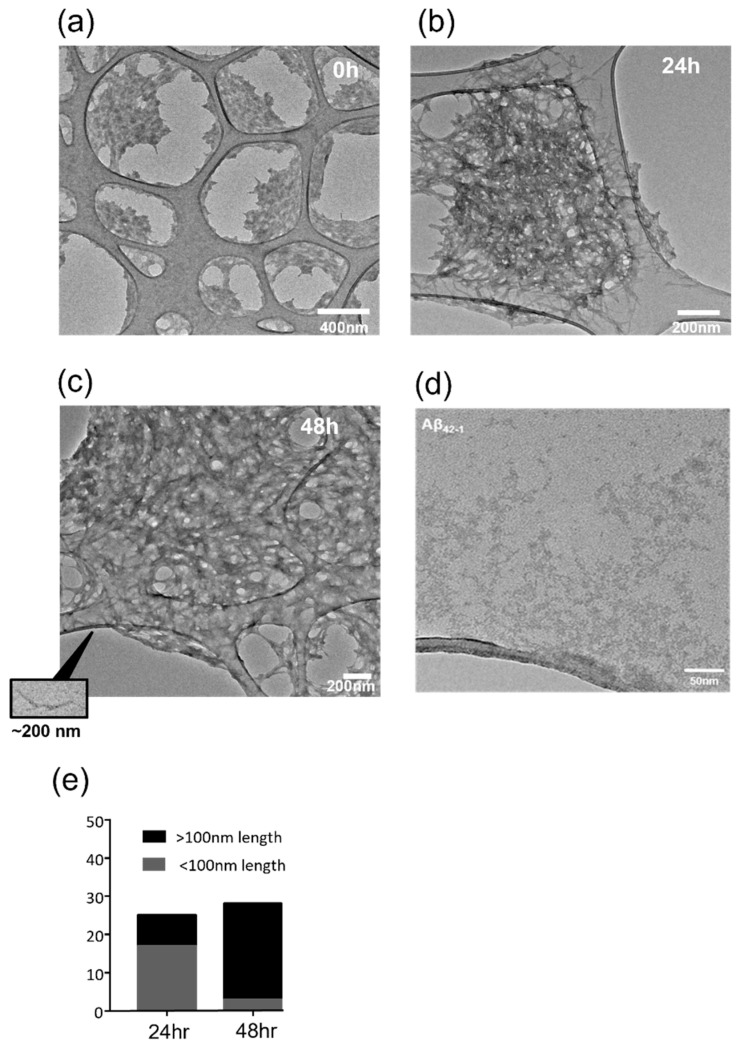
**Temporal profile of Aβ_1-42_ fibrillization**. Transmission electron microscopy (TEM) was used to monitor Aβ_1-42_ fibrillization. Representative images (from n = 22) using a NH_4_OH dissolution protocol (see Methods) showed that (**a**) at 0 h, the peptide was largely not aggregated, (**b**) at 24 h, peptide fibrillization had started (protofibrils <100 nm) and (**c**) by 48 h, the peptide aggregated into larger protofibrils (>100 nm in length); (**d**) as a comparison, Aβ_42-1_ was analysed, revealing that it did not aggregate into protofibrils and it remained as unaggregated clumps. (**e**) Image J analysis of protofibril length revealed that by 48 h, the majority of the protofibrils were over 100 nm in length, and at 24 h with the NH_4_OH protocol the predominant form was <100 nm (n = 22–28).

**Figure 2 pharmaceuticals-15-01459-f002:**
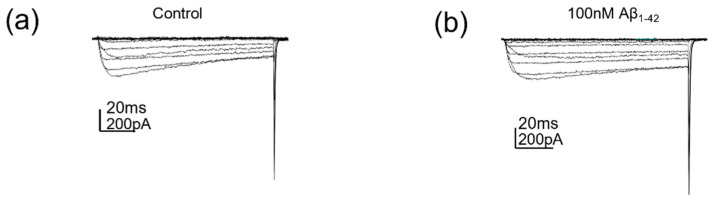
**The 100 nM protofibrillar Aβ_1-42_ did not affect Cav2.2 currents**. (**a**) Cav2.2 currents evoked by voltage steps (−70 to +50 mV) following control (PBS) treatment and (**b**) Cav2.2 currents evoked by voltage steps (−70 to +50 mV) following 100 nM protofibrillar Aβ_1-42_ treatment.

**Figure 3 pharmaceuticals-15-01459-f003:**
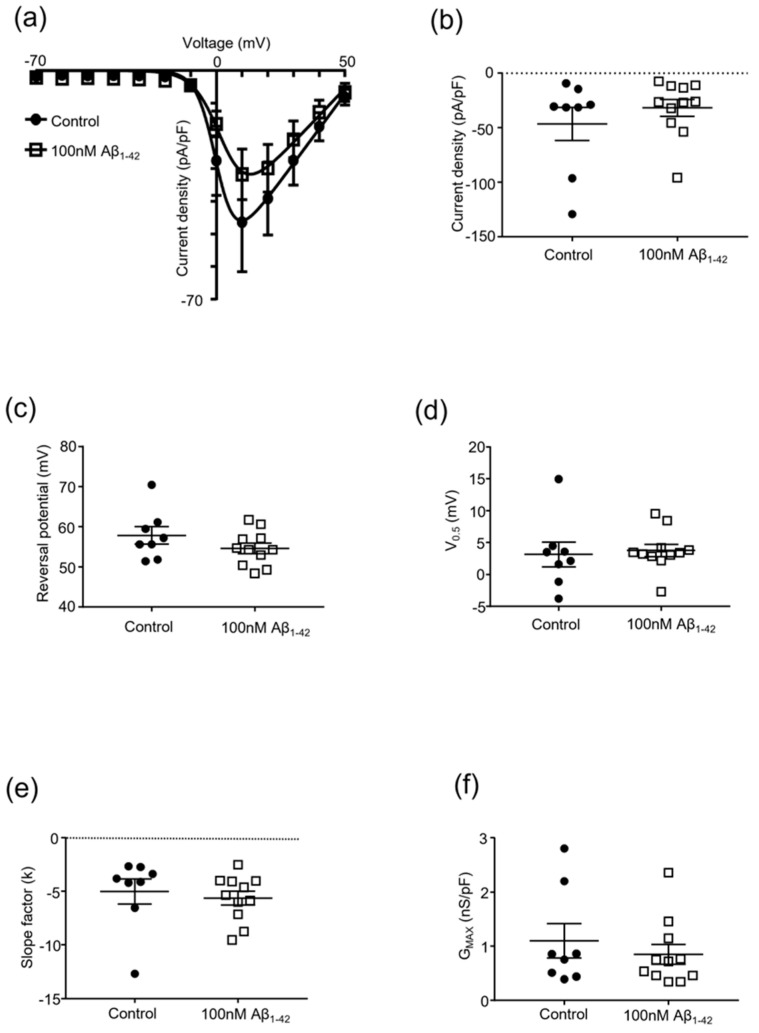
**Effect of protofibrillar 100 nM Aβ_1-42_ on Cav2.2 kinetics**. Whole-cell patch clamp popul-tion experiments investigating the effect of protofibrillar 100 nM Aβ_1-42_ vs. vehicle control (PBS) on Cav2.2 channel kinetics of Cav2.2 stably transfected HEK293 cells; mean ± SEM, n = 8–11 throughout. (**a**) Current density–voltage (−70 to +50 mV) relationship; (**b**) current density of Cav2.2 channels at +10 mV; (**c**) reversal potential of Cav2.2 current; (**d**) half-activation voltage of Cav2.2 channels; (**e**) changes in the slope factor (k) of Cav2.2 current; and (**f**) maximal conductance of Cav2.2 channels.

**Figure 4 pharmaceuticals-15-01459-f004:**
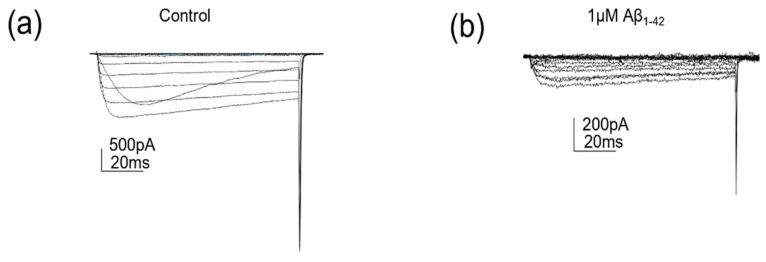
**1 µM protofibrillar Aβ_1-42_ reduced Cav2.2 currents**. (**a**) Cav2.2 currents evoked by voltage steps (−70 to +50 mV) following control (PBS) treatment and (**b**) Cav2.2 currents evoked by voltage steps (−70 to +50 mV) following 1 µM protofibrillar Aβ_1-42_ treatment.

**Figure 5 pharmaceuticals-15-01459-f005:**
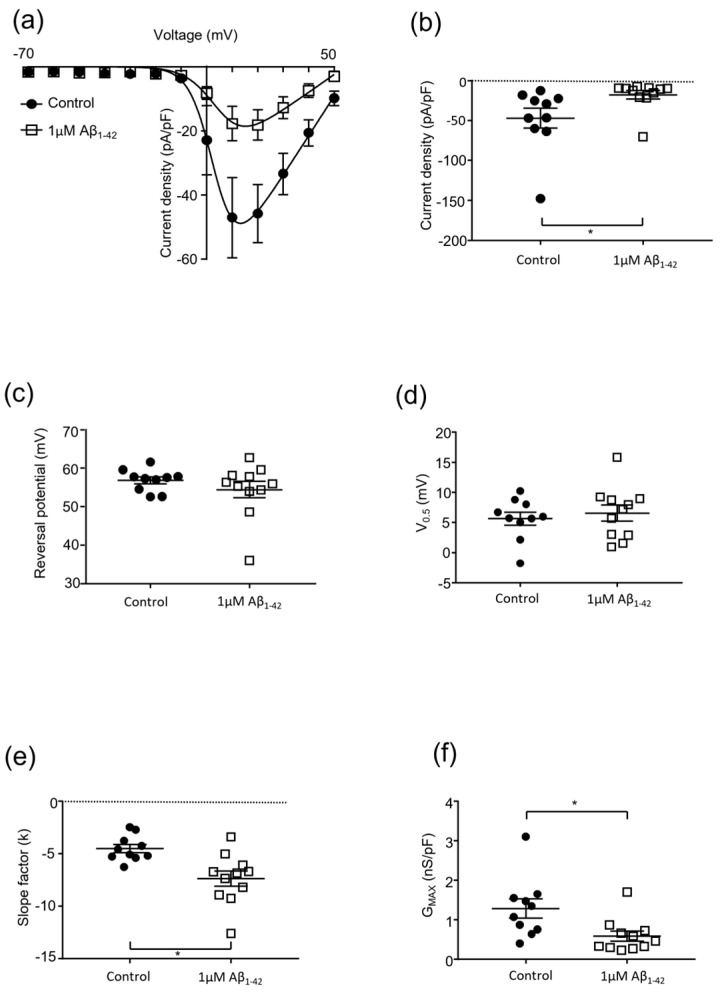
**Effect of protofibrillar 1 mM Aβ_1-42_ on Cav2.2 kinetics**. Whole-cell patch clamp population experiments investigating the effect of protofibrillar 1 µM Aβ_1-42_ vs. vehicle control (PBS) on Cav2.2 channel kinetics of Cav2.2 stably transfected HEK293 cells; mean ± SEM, n = 15 throughout. (**a**) Current density–voltage (−70 to +50 mV) relationship; (**b**) current density of Cav2.2 channels at +10 mV; (**c**) reversal potential of Cav2.2 current; (**d**) half-activation voltage of Cav2.2 channels; (**e**) changes in the slope factor (k) of Cav2.2 current; and (**f**) maximal conductance of Cav2.2 channels. * = *p* < 0.05, two-tailed unpaired *t*-test.

## Data Availability

Data is contained within the article.

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
