# Peer review of "Protofibrillar Amyloid Beta Modulation of Recombinant hCaV2.2 (N-Type) Voltage-Gated Channels"

_pharmaceuticals, 2022, doi:10.3390/ph15121459_

Round 1

Reviewer 1 Report

In this short communication Kaisis et al conducted electrophysiology study to determine the effects of protofibrilar Aβ42 on Cav2.2 channel.  They have shown that 100nM protofibrilar Aβ42 do not modulate Cav2.2, but higher concentration (1uM) alters the biophysical properties of this channel. The results are very interesting, but several concerns need to be resolved before publication:

1.       It is very difficult to understand how Aβ42 was prepared (line 222-237). Did the author first induced aggregation and measured concentration of protofibrils (100nM and 1uM) or took two different concentrations of monomers and induces aggregation? If second procedure is followed, 1uM aggregates supposed to be different protofibril/fibril species than 100nm concentration. There is no comparison with monomers even though they mentioned monomeric (control) Aβ was not sonicated (line 227). Did the authors use reverse peptide (Line 236)? In which experiment the ratio was used?

2.       The control condition was not considered appropriately in cell culture experiments. The author mentioned that PBS was used as control (Line 91). However, they treated Aβ with 1% NH4OH to make film, then diluted with 10mM NaOH, and finally used 2% SDS in stock solution. Only PBS cannot be considered as a good control for this Aβ. A proper control need to be included to eliminate the effects of NH4OH, NaOH, and SDS on Cav2.2 channel properties.

Minor

Please clarify 5mM, 100mM, and 400mM stock during Aβ preparation.

Reviewer 2 Report

Kaisis et al. investigated the effect of protofibril Aβ1-42 on Cav2.2 channel modulation. They found out that only the high concentration (1um) showed the modulatory effects. This is very straightforward study with clear results, however, some critical concerns must be addressed before a positive comment can be made.

1 both cell line and primary murine microglia should be employed, instead of only using HEK 293 cell. Why did authors only focus on the HEK293?

2 it was reported that Aβ protofibrils are toxic to neurons. Will the high concentration of protofibril Aβ1-42 affect the cellular viability? The cell viability should be evaluated after treatment with protofibril Aβ1-42.

3 will this Cav2.2 channel modulation be recovered after the effect of protofibril Aβ1-42 withdrew?

4 Authors showed the effects of different concentration separately, and I suggest authors to combine some data in figures.

Round 2

Reviewer 1 Report

To avoid confusion, please write Ab1-42 and Ab42-1 instead of Ab1-42/Ab42-1, which seems to mean a ratio.  

Still I am not clear why  100uM Ab was prepared from 5nM and incubated for 48h. Next, 400uM was prepared and incubated for 24h, and then diluted to 100uM and incubated for 24h.

If those preparations are for different experimental purposes, please clarify.

Author Response

We thank the reviewer for the appraisal of our revised manuscript and their valuable insight. We have further commented on their below to consider their additional two points raised:

1) We apologise for the confusion in the use of our terminology and have now corrected the use of Ab1-42 and Ab42-1, to make it clear it is a single species rather than a ratio we are referring to.

2)  We apologise for the continuned confusion over our amyloid beta preparation which we had sought to clarify previously. To enhance the clarity of the methodolgy we have restructure this section. We hope that this now provides an accessible step by step guide to our preparation and how this was used for our cellular experiments. This process is run over 48hrs which starts with the 5mM (the reviewer mentions 5nM, but the manuscript states 5mM) stock and results in 100µM which is then used as our working concentration. This is a two step process, firstly diluting the 5mM to 400µM (incubated for 24hrs) and then followed by a subsequent dilution from 400µM to 100µM (incubated for 24 hrs). We hope this revision of the text makes the methodology clear and reproducible for readers. 

Reviewer 2 Report

I have no more comments on this manuscript. Authors addressed all my concerns well. 

Author Response

We thank the reviewer for their appraisal of our responses to their valuable insight.